# Effects of Polypropylene Fibers on the Frost Resistance of Natural Sand Concrete and Machine-Made Sand Concrete

**DOI:** 10.3390/polym14194054

**Published:** 2022-09-27

**Authors:** Yan Tan, Junyu Long, Wei Xiong, Xingxiang Chen, Ben Zhao

**Affiliations:** 1Department of Architecture and Environment, Hubei University of Technology, Wuhan 430068, China; 2Department of Architecture and Engineering, Wuhan City Polytechnic, Wuhan 430064, China

**Keywords:** polypropylene fiber mechanism sand concrete, frost resistance, freeze–thaw damage model, RSM strength model

## Abstract

In order to study the effect of polypropylene fibers on the frost resistance of natural sand and machine-made sand concrete, polypropylene fibers (PPF) of different volumes and lengths were mixed into natural sand and machine-made sand concrete, respectively. The freeze–thaw cycle test was carried out on polypropylene-fiber-impregnated natural sand concrete (PFNSC) and polypropylene-fiber-impregnated manufactured sand concrete (PFMSC), respectively, and the apparent structural changes before and after freezing and thawing were observed. Its strength damage was analyzed. A freeze–thaw damage model and a response surface model (RSM) were established used to analyze the antifreeze performance of PFMSC, and the effects of the fiber content, fiber length, and freeze–thaw times on the antifreeze performance of PFMSC were studied. The results show that with the increase in the number of freeze–thaw cycles, the apparent structures of the PFMSC gradually deteriorated, the strength decreased, and the degree of freeze–thaw damage increased. According to the strength damage model, the optimum volume of PPF for the PFNSC specimens is 1.2%, and the optimum volume of PPF for the PFMSC specimens is 1.0%. According to the prediction of RSM, PFNSC can maintain good antifreeze performance within 105 freeze–thaw cycles, and when the PPF length is 11.8 mm, the antifreeze performance of PFNSC reaches the maximum, its maximum compressive strength value is 33.8 MPa, and the split tensile strength value is 3.1 MPa; PFMSC can maintain a good antifreeze performance within 96 freeze–thaw cycles. When the length of PPF is 9.1 mm, the antifreeze performance of PFMSC reaches the maximum, its maximum compressive strength value is 45.8 MPa, and its split tensile strength value is 3.2 MPa. The predicted values are in good agreement with the measured values, and the model has high reliability.

## 1. Introduction

Durability is an important indicator for measuring the ability of a material to resist the long-term damage of both itself and the natural environment [1,2,3,4]. Freeze–thaw damage in cold regions has an important impact on the durability of concrete structures. Freeze–thaw damage to concrete in alpine regions of China is a common issue, and it poses a huge threat to the safety of building use and economic and environmental protection. Frost resistance is particularly important. The main reason for the freeze–thaw damage of concrete is that the water in the concrete pores, under the action of alternating dry and wet and freeze–thaw cycles, forms fatigue stress from the combined action of ice expansion pressure and osmotic pressure, which causes the concrete to produce denudation damage from the surface to the inside, thereby reducing the strength of the concrete [5,6,7]. Research on the antifreeze performance of concrete structures under low-temperature freeze–thaw cycle environment can not only reveal potential dangers and avoid major safety accidents but also provide a basis for durability design, testing, and reinforcement of concrete structures in low-temperature freeze–thaw environments. Improving the frost resistance of concrete has become a hot topic of research at home and abroad [8,9,10].

Compared with ordinary concrete, adding fiber is one of the methods that can effectively reduce the brittleness and improve the toughness of concrete [11,12,13,14]. Fiber materials have been widely studied by scholars at home and abroad. The results show that fiber structures have the characteristics of strong plasticity, high toughness, and strong adhesion to concrete, and adding fibers into concrete can offset the internal stress and play an effective role in frost resistance [15,16,17,18]. The different types of fibers have different effects on the frost resistance of concrete. Among them, not only is polypropylene fiber (PPF) small in diameter, low in quality, low in cost, and good at self-dispersion, but it also has the characteristics of inhibiting concrete plastic cracking, preventing crack propagation, limiting matrix damage, improving the long-term working performance of concrete structures, and improving structural durability [19,20,21]. In the process of freezing, the increase in the elastic modulus of PPF can effectively offset the ice expansion force, and the decrease in the elastic modulus of PPF during the thawing process helps it to release the accumulated expansion energy. Therefore, the change in the elastic modulus of PPF is enhanced as a whole. The strain capacity of concrete in the freeze–thaw environment can effectively inhibit the frost heave cracking of concrete. Different lengths and dosages of PPF have varying degrees of influence on the tensile properties, fatigue resistance, and wear resistance of concrete [22,23,24]. The performance of concrete has been improved in all aspects [25]. Cheng Hongqiang et al. [26] conducted a freeze–thaw damage test of polypropylene fiber concrete. Under the action of freeze–thaw cycles, the damage of polypropylene fiber concrete continued to accumulate, and the mass loss rate and split tensile strength continued to decrease. The splitting tensile strength has been continuously improved. Chen Liuzhuo et al. [27] studied the frost resistance of ordinary concrete by adding steel fibers and polypropylene fibers. The results show that controlling the amount and length of fibers can effectively improve tensile strength and flexural strength. Durability is enhanced, and polypropylene fibers are more damage-resistant than steel fibers. Salehi Parisa et al. [28] studied the effect of PPF on the mechanical properties and durability of reinforced lightweight concrete through the mixed design of three volume dosages (0%, 0.5%, and 1%) of PPF, and the results show that the incorporation of fibers can improve compressive, tensile, and flexural strength, and it also reduces the water absorption and permeability of concrete. Liu Bo et al. [29] analyzed and summarized the influence of polypropylene fibers on the mechanical properties and durability of concrete and proved that the addition of polypropylene fibers can make concrete mechanically resistant to compression, tension, bending, and impact. In terms of performance and durability, including frost resistance, impermeability, and carbonation resistance, it is superior to ordinary concrete.

Due to the over-exploitation of natural sand, natural sand resources are depleted, which seriously affects the ecological environment, and finding new alternative sources has become a hot topic. Machine-made sand is artificially mined industrial waste slag, tailings, etc., which are mechanically crushed and screened into rocks with a particle size of less than 4.75 mm [30,31,32]. The mud powder in natural sand has an adverse effect on the working performance, volume stability, and durability of concrete, while the fine-grained stone powder in machine-made sand can improve the gradation of machine-made sand and fill the pores of the particles. A large number of studies have shown that an appropriate amount of stone powder can improve the workability of concrete and improve the strength and durability of machine-made sand concrete [33,34,35]. Replacing natural sand with manufactured sand has different degrees of influence on the mechanical properties of concrete. Ding et al. [36] prepared manufactured sand concrete with a compressive strength of higher than 50 MPa, and some scholars have prepared ultra-high-pressure concrete through different treatment methods. The compressive strength of high-performance concrete exceeds 140 MPa [37,38,39,40]. Bonavetti and Irassar [41] studied the effect of machine-made sand and gravel powder on the properties of mortar. With an increase in age and in the replacement rate of machine-made sand, the compressive strength of concrete showed an increasing trend. Donza et al. [42] studied the effects of several different materials of manufactured sand and natural sand on the properties of high-strength concrete. The results showed that the mechanical properties of high-strength concrete were better, due to the mechanical occlusion of the manufactured sand. Zeghichil et al. [43] studied the effect of natural sand and multi-angled machine-made sand on the working performance and mechanical properties of self-compacting concrete. Multi-angled machine-made sand requires more water to achieve the same working performance, but the compressive strength of the concrete is lower than that of regular concrete. The flexural strength is higher than that of concrete prepared from natural sand.

Regarding the application of fiber materials in machine-made sand concrete, Kandasamy and Murugesan [44] studied the compressive strength and split tensile strength of machine-made sand concrete and concrete mixed with plastic fibers. The results show that the addition of plastic fibers can effectively improve the mechanical properties and durability of manufactured sand concrete. Some scholars have improved the shortcomings of easy cracking and poor frost resistance by admixing PPF and mineral admixtures in desert sand concrete [45]. In view of the excellent performance of polypropylene fibers in concrete compared with other fibers, there are relatively few studies on the freeze–thaw cycle of polypropylene fiber concrete in which continuous graded machine-made sand replaces 100% of natural fine aggregates.

In this paper, by incorporating polypropylene fibers of different lengths and volumes, natural sand concrete and manufactured sand concrete with a replacement rate of 100% were used as the objects to study the change law of the durability of PFNSC and PFMSC under freeze–thaw cycles. A fitting model corresponding to the fiber lifting rate and contribution rate [46] was established to quantitatively characterize the influence of PPF. At the same time, based on learning from the classical combination model of damage [47,48,49,50,51], the damage theory under freeze–thaw cycles was used [52,53]. A freeze–thaw damage model and a response surface model were established to analyze the frost resistance of PFNSC and PFMSC, and the effects of fiber content, fiber length, and freeze–thaw times on the frost resistances of PFNSC and PFMSC were studied. The PFNSC and PFMSC specimens were compared.

## 2. Materials and Methods

### 2.1. Raw Materials and Mix Ratio

The cementitious materials used in this test are cement and grade I fly ash, the cement type is standard P.O. 42.5 grade ordinary cement, and all the cement indexes meet the requirements of GB175-2007 “General Portland Cement”. Among the materials, fly ash complies with the requirements of the standard GB/T 1596-2005 “fly ash used in cement-based concrete”, its apparent density is 2500 kg/m^3^, the specific surface area is 455.2 m^2^/kg, and it is a 30 mm continuous graded natural aggregate; fine aggregates are natural river sand (NRS) and machine-made sand(MS) in zone II, for which the fineness modulus is 2.9, the apparent density of natural river sand is 2650 kg/m^3^, and the apparent density of manufactured sand is 2610 kg/m^3^. According to the standard GB/T14684-2011 “Sand for Construction”, the MB value and stone powder content of machine-made sand meet the requirements of Class II machine-made sand. The superplasticizer is polycarboxylate, and the gradation curve is shown in Figure 1. The physical map and microscopic topography of the polypropylene fibers are shown in Figure 2a,b, and their performance indicators are shown in Table 1.

It can be seen from Figure 1 that the surface characteristics of the gradations of machine-made sand are that the occupancy of the middle section is higher than the occupancy of both ends, that is, “large in the middle and small at both ends”, and the degree of gradation continuity is poorer than that of natural fine aggregate. In this test, the benchmark concrete strength grade is C30, and natural river sand and machine-made sand are used as fine aggregates. The mix ratio of fiber concrete is shown in Table 2. The percentage of polypropylene fibers should be controlled at about 1.0% and not more than 1.2% [54]. In this paper, the volume content of PPF is 0%, 1.0%, and 1.2%. PFNSC a–b and PFMSC a–b are 12 mm and 19 mm (“a” is the volume content of PPF and “b” is the length of PPF), respectively.

### 2.2. Test Equipment and Test Methods

A total of 14 sets of specimens were designed for the test, all of which conform to GB/T 50082-2009 “Standards for Test Methods for Long-term Performance and Durability of Ordinary Concrete”.

The compressive test uses a DYE-2000A microcomputer servo pressure-testing machine (produced by Cangzhou Zhulong Engineering Instrument Co. Ltd., Cangzhou, China), as shown in Figure 3a, and a cube with a size of 100 mm × 100 mm × 100 mm to measure the compressive strength test. The test loading speed is 0.5 MPa/s, and the compressive strength correction coefficient is 0.95. After the test piece reaches the test age, it is taken out of the curing room. The surface of the test piece and the upper and lower bearing plates are wiped, and a check is made for whether the center of the test piece is aligned with the center of the lower pressure plate of the testing machine. After the inspection is completed and confirmed to be correct, the compressive strength test is carried out on the testing machine. The loading speed of the press is manually controlled. When the specimen is close to failure and begins to deform sharply, we stop adjusting the test throttle until failure and record the failure load. The arithmetic mean of the measured values of 3 test pieces is taken as the strength value of the test piece.

The split tensile test uses a DYE-2000A microcomputer servo pressure-testing machine (produced by Cangzhou Zhulong Engineering Instrument Co., Ltd., Cangzhou, China), as shown in Figure 3b. Cubes with dimensions of 100 mm × 100 mm × 100 mm were tested for split tensile strength. The test loading speed is 0.08 MPa/s. After the specimen is taken out, the dry and wet state of the specimen are kept unchanged, and measurements are taken to determine whether the size of the specimen meets the requirements. A line is drawn in the middle of the side of the specimen during forming to determine the position of the splitting surface, the lower cushion and gasket are placed in the center of the lower plate, and the specimen, upper gasket, and cushion are placed in sequence on the lower gasket. The contact busbar of the gasket is aligned with the load contact line on the specimen accurately, and the contact between the upper pressure plate and the cushion is adjusted. After checking the adjustment and correctness, turn on the split tensile testing machine. When the specimen is close to failure or deformation, we stop adjusting the test throttle until failure, and the value is recorded. Three test blocks are set for each group of mix ratios, and the value method is the same as that of the compressive strength value.The freeze–thaw test is performed using a TDR-III(produced by Hebei Huawei Co., Ltd., Cangzhou, China), automatic rapid freeze–thaw test machine, as shown in Figure 3c, with a cylindrical block with a size of 100 mm × 100 mm × 400 mm, and a freeze–thaw cycle test. This is carried out according to the “slow freezing method” in GB/T 50082-2009 “Standard for Long-term Performance and Durability Test of Ordinary Concrete”. After the test piece reaches the test age, it is taken out from the curing room four days in advance and immersed in water with a temperature of 15~20 °C. After soaking, the test piece is taken out to dry the surface moisture, and its initial mass is weighed. A total of 150 freeze–thaw cycles were carried out in this test, and measurements after every 50 freeze–thaw cycles were used to collect relevant frost resistance durability parameters. The collected contents included: concrete specimen quality, compressive strength, and flexural strength.

## 3. Test Results and Discussion

### 3.1. Strength Analysis after Freeze–Thaw Cycle

The PFNSC and PFMSC specimens were set up in the experimental group and the control group with different fiber content, fiber length, and manufactured sand content. They were divided into 14 groups, and each group was set with three specimens, a total of 42 test blocks. The compressive strength, split tensile strength, and quality changes after 0, 50, 100, and 150 freeze–thaw cycles are shown in Table 3.

Using the results of the orthogonal test, the compressive strengths of the PFNSC and PFMSC specimens with different freezing and thawing times and fiber lengths under different polypropylene contents were generated using origin 2021 software (produced by OriginLab Co. Ltd., Northampton, MA, USA), as shown in Figure 4. It can be seen from Figure 4a,b that, with the increase in the number of freeze–thaw cycles, different PPF volume contents in PFMSC specimens had a greater impact on the compressive strengths of the specimens. With an increase in the number of freeze–thaw cycles, the strength of the specimen with a volume of 1.0% decreased rapidly. The strength of the concrete specimen with a volume of PPF of 1.2% was higher than that of the specimen with a 1.0% volume of PPF under different freeze–thaw cycles. The 1.0% concrete specimen shows that the 1.2% fiber volume content results in better frost resistance than the 1.0% fiber volume content. After 50–150 freeze–thaw cycles, the compressive strength of the specimen decreased first, then increased, and then decreased with the increase in fiber length, and the deceleration rate reached the maximum when the fiber length was 12 mm. The comparison of Figure 4a–d shows that with the increase in freeze–thaw times, the average reduction rate for the compressive strengths of the PFNSC and PFMSC specimens is close. The average reduction in the compressive strengths of PFNSC specimens is 26.4%; the average reduction in the compressive strengths of the PFMSC specimens is 26.2%. This shows that the frost resistance of the PFNSC and PFMSC specimens is equivalent.

Figure 5 shows the split tensile strength of PFNSC and PFMSC specimens with different freezing and thawing times and fiber lengths, respectively, for different polypropylene contents. It can be seen from Figure 5a,b that after 50–150 freeze–thaw cycles of the PFMSC specimens, the split tensile strength of the concrete is significantly lower than that of the control group without PPF. The split tensile strength decreases with the increase in fiber length, showing a trend of increasing first and then decreasing. For example, the split tensile strength of the PFMSC specimen with 50 freeze–thaw cycles and 1% PPF volume content is 1.47 times that of the plain concrete without a PPF resemblance. When the volume content of PPF is constant, the relative split tensile strengths of the concrete specimens gradually decrease with the increase in the number of freeze–thaw cycles, but after 100 freeze–thaw cycles, the split tensile strengths of PFMSC specimens decreases. Under the condition of constant fiber content, the relative split tensile strength decreased with the increase in fiber length, with the strength first increasing and then decreasing. By comparing Figure 5a–d, it is found that with the increase in freezing and thawing times, the average reduction rate of the split tensile strengths of PFNSC specimens is slightly higher than that of PFMSC specimens, indicating that the replacement of natural sand concrete by machine-made sand concrete can effectively enhance the durability of concrete. This is because appropriate stone powder in the PFMSC specimen can improve the grading of machine-made sand and fill the pores of particles, which improves the durability of concrete. After 50–150 freeze–thaw cycles, the average reduction in the splitting tensile strengths of PFNSC specimens was 32.5%; the average reduction in splitting tensile strengths of the PFMSC specimens was 23.2%. This shows that the antifreeze performance of PFMSC is better than that of the PFNSC specimens.

The test results show that freeze–thaw cycles have a great influence on the compressive strengths and split tensile strengths of the PFNSC and PFMSC specimens. The addition improved the antifreeze properties of PFNSC and PFMSC. Different PPF volume contents and PPF lengths also have a certain degree of influence on the mechanical properties of concrete. It can be seen that, after 0–150 freeze–thaw cycles, when the volume of PPF is the same, the durability energies of PFMSC and PFNSC increase first and then decrease with the increase in the length of PPF; when the length of PPF is 6–12 mm, the mechanical properties of PFMSC and PFNSC increase with the increase in fiber volume content. When the PPF length is 19 mm, the mechanical properties of PFMSC and PFNSC decrease with the increase in fiber volume content.

Figure 6 shows the apparent state of the PFMSC specimen after 0, 50, 100, and 150 freeze–thaw cycles and the surface macro-state of the binarized concrete specimen under the same variables. It can be seen from Figure 6 that after 0 freeze–thaw cycles, there is a small area of cement material shedding on the surface of the specimen; that is, the black area marked in red is the area where a small part of the surface of the specimen has fallen off the cement material. With the increase in the number of freeze–thaw cycles, the quality of the PFMSC specimen continued to decrease, and the coarse aggregate inside the PFMSC specimen was gradually exposed. After 50 freeze–thaw cycles, a small area of the surface of the specimen had fallen off, and the mass loss was approximately 0.32%. After 100 freeze–thaw cycles, much of the surface of the test piece had fallen off, large coarse aggregate particles were exposed, and the mass loss was approximately 1.85%. When the freeze–thaw cycles reached 150, the surface of the specimen showed a covered black area, indicating that the surface of the concrete specimen was almost completely peeled off under 150 freeze–thaw cycles, the aggregate under the surface was completely exposed, and the mass loss was approximately 4.5%. With the increase in the number of freeze–thaw cycles, the surface peeling degree of the specimens gradually deepened, the strength damage of the PFMSC specimens was serious, and the mass loss was large.

### 3.2. Freeze–Thaw Injury of PFNSC and PFMSC under Strength Evaluation Index

In order to consider the change in freeze–thaw cycles and the decay law of concrete mechanical properties, the relationship between the compressive and split tensile strengths of the freeze–thaw cycle damage degree under different polypropylene fiber volume contents and lengths was analyzed. Based on the data changes in compressive strength and splitting tensile strength, according to “Concrete Damage Mechanics”, the degree of freeze–thaw cycle damage of concrete is represented by *D*, and *D*_c_ and *D*_t_ are defined as the degree of freeze–thaw cycle damage under compression and splitting resistance, respectively, which correspond to strength index damage. They are calculated by Formulas (1) and (2).
(1)$$D_{c,m}=1-\frac{f_{c,n}}{f_{c,0}}$$
(2)$$D_{t,m}=1-\frac{f_{t,n}}{f_{t,0}}$$

In the formulas:

$D_{c,m}$—Compressive strength damage variable (%); $D_{t,m}$—Split tensile strength damage variable (%);

$f_{c,n}$—Compressive strength (MPa) under corresponding freeze–thaw times; $f_{t,n}$—Splitting strength (MPa) under corresponding freeze–thaw times.

In the formula, ‘c’ indicates that the specimen is a compression specimen, ‘t’ indicates that the specimen is a split-pull specimen, ‘n’ is the number of freeze–thaw cycles, and ‘m’ is the content of manufactured sand. Through Formulas (1) and (2), the strength damage model of the polynomial function under the freeze–thaw cycles was established, and the strength damage of the PFNSC and PFMSC specimens was predicted using the model, as shown in Figure 7a–d under different freeze–thaw cycles.

It can be seen from Figure 7 that the extents of freeze–thaw damage of the compressive strengths and split tensile strengths of the PFNSC and PFMSC specimens increase with the increase in the number of freeze–thaw cycles. From Figure 7a,b on tensile strength damage, it can be seen that compared with PFNSC, the compressive strength of the PFMSC specimen is more damaged, because the shape of the machine-made sand is sharp and rough and has many edges and corners. After a certain number of freezing and thawing cycles, the machine-made sand will accelerate the material falling off. After PFNSC freeze–thaw cycles, the freeze–thaw damage rate without PPF is greater than that with PPF. This is because the polypropylene fibers are constrained in the concrete and bear part of the damage caused by the freeze–thaw cycles inside the specimen. With the effect of force, the freeze–thaw damage rate of PFMSC without PPF is lower than the freeze–thaw damage rate of the PPF-doped specimens, and generally, the longer the length of PPF, the greater the amount of freeze–thaw damage. This is due to the mechanism of the sand particles. The influence of shape characteristics on the compressive strengths of PFMSC specimens is greater than that of fibers, and the longer the fibers, the larger the contact area with machine-made sand will be. It can be seen from Figure 7c,d that the split tensile strength damage rate is significantly reduced by PPF. The freeze–thaw damage rate of the specimens without PPF is greater than the freeze–thaw damage rate of the specimens with PPF. Longitudinal pressure inside the specimen can improve the durability of concrete; PFNSC specimens have the same length of PPF, and the splitting tensile strength of the specimen when the volume content of PPF is 1.2% is less than that of the specimen when the volume content of PPF is 1%. The freeze–thaw damage rate of PFMSC is due to the bending of PPF when the inside of the specimen is greatly constrained, which increases the contact area and improves the anti-splitting ability; PFMSC specimens have the same length of PPF, and the volume content of PPF is 1.2%. The freeze–thaw damage rate of the specimen is greater than the freeze–thaw damage rate of the specimen with a volume content of 1% PPF. The mechanism is that the influence of the shape characteristics of the machine-made sand particles on the splitting tensile strength of PFMSC is greater than that of the fiber inside the specimen. The fitting function of the freeze–thaw damage rate is shown in Table 4 and Table 5. It can be seen from Table 4 and Table 5 that the fitted regression curve and the observed value of R^2^ are both greater than 0.95, indicating that the fitted model is highly reliable.

## 4. Model of the Composite Factor RSM Intensity

The RSM uses mathematical and statistical methods to model and analyze problems affected by multiple variables, with the ultimate goal of optimizing the response value [55,56]. Box and Wilson first proposed the response surface method. At that time, the research on the response surface method was limited to how to obtain an explicit function using statistical methods to approximate a complex implicit function. Fang et al. [57] used the D-optimal design and a first-order response surface model to predict the dynamic response and damage identification of intact and damaged systems, and they also used numerical examples, reinforced concrete frame model tests, and I-40 real bridge test results to verify the effectiveness of the proposed method. Using the response surface method (RSM) to model the relationship between factors and levels can better analyze the accuracy, significance, and reliability of the experimental data. Zong Zhouhong et al. [58] used the center composite design (CCD) method and the response surface model to complete the finite element model correction of the Baishi Bridge, and they proved that the bridge finite element model correction based on the response surface method has a higher accuracy. Many scholars at home and abroad have established relational models through the response surface method and have obtained correspondingly optimal results [59,60,61,62].

The damage degree of PFNSC and PFMSC under different conditions was quantitatively analyzed via freeze–thaw damage. This showed that when the length of PPF is the same and the volume content of PPF is 1.2, the antifreeze performance of PFNSC is optimal; when the volume content of PPF is 1.0, the antifreeze performance of PFMSC is optimal. In order to better study the effect of PPF length and the number of freeze–thaw cycles on antifreeze performance, that is, taking the PPF length and the number of freeze–thaw cycles as two factors, a response surface strength model was established, and design-Expert software was used to conduct a multivariate analysis of the experimental data. Regression analysis was performed to establish the fitting model of Equations (3)–(6). Equations (3)–(6) were analyzed via variance analysis. The regression coefficient value *R*^2^ was used to test the reliability of the model. When the regression coefficient value *R*^2^ was close to 1, the model reliability was high.
(3)$$f_{c0}=63.91-2.63A+0.13B+2.95C+1.59AB-0.81AC-2.36BC+2.13A^{2}-8.27B^{2}+5.4C^{2}\;R^{2}=0.8851$$
(4)$$f_{c100}=46.57-2.56A-0.46B-0.75C-0.97AB-1.24BC+3.25A^{2}-2.8B^{2}-7.04C^{2}\;R^{2}=0.9050$$
(5)$$f_{t0}=2.4-0.47A-0.083B+0.37C+0.027AB-0.1BC+0.085A^{2}-0.17B^{2}+0.22C^{2}\;R^{2}=0.9368$$
(6)$$f_{t100}=3.57-0.36A+0.011B+0.28C+0.12AB+\left(2.85E-003\right)BC-0.27A^{2}-0.018B^{2}-0.59C^{2}\;R^{2}=0.9248$$

In the formula:

$f_{c0}$—PFNSC compressive strength (MPa); $f_{c100}$—PFMSC compressive strength (MPa);

$f_{t0}$ —PFNSC tensile strength (MPa); $f_{t100}$—PFMSC split tensile strength (MPa);

A—Freeze–thaw times (*n*); B—Polypropylene fiber length (mm); C—Polypropylene fiber volume content (%).

It can be seen from the fitting model that the values obtained by the fitting equation are all close to 1 and that the value of the coefficient of variation obtained by the model is very small, indicating that the correlation between the model and the test data is significant and the degree of fitting is high, which indicates the model can better analyze and predict freezing conditions. The effects of the PPF length and the number of freeze–thaw cycles on the antifreeze properties of PFNSC and PFMSC specimens within 150 thaw cycles are found. Figure 8 and Figure 9 show the optimal three-dimensional response surfaces and contour maps of the compressive and split tensile strengths of the PFNSC specimen and the PFMSC specimen obtained, respectively, based on the RSM model within 150 freeze–thaw cycles.

When the volume content of PPF is 1.2%, the PFNSC contour map and 3D response surface are as shown in Figure 8. Through the analysis of Figure 8a,b, it can be seen from the contour map that the green on both sides gradually changes to orange in the center and that the center is the region with the highest responsivity, indicating that the material content corresponding to this region is the optimal doping value. In the three-dimensional response surface graph, the variation trend of the R value along factor B is greater than that of factor A, indicating that factor B (number of freeze–thaw cycles) has a greater impact on the R value than factor A (fiber length). When the number of freeze–thaw cycles increased from 50 to 150, the compressive strength and split tensile strength of PFNSC decreased by 27.5 MPa and 1.8 MPa, respectively; that is, the strength of the specimen decreased with the increase in the number of freeze–thaw cycles. When the fiber increased from 6 mm to 19 mm (1.8 MPa), the strength of the specimen showed a trend of first increase and then decrease, and when the fiber length was approximately 10 mm, the tensile strength reached the maximum. As can be seen from Figure 8c,d, the green on the top of the contour map gradually changes to the red on the bottom. The region below the center is the region with the highest response, and the corresponding polypropylene fiber length and the number of freeze–thaw cycles reach optimum values. The density of the contour lines on the ordinate in the split tensile strength contour diagram is greater than the density of the contour lines on the abscissa; that is, when the fiber volume content is 1.2%, the effect of fiber length on the split tensile strength is greater than that of freezing. The effect of the number of melting cycles is seen.

As shown in Figure 10, when the volume content of the PFNSC fibers is 1.2%, the response surface model can predict that when the number of freeze–thaw cycles is 105 and the fiber length is 11.8 mm, the compressive strength and splitting tensile strength of PFNSC are both at maximum values of 33.8 MPa and 3.1 MPa, respectively, indicating that PFNSC can maintain a good antifreeze performance within 105 freeze–thaw cycles. The desirability is 0.975, indicating that the model has high prediction reliability.

As shown in Figure 11, when the volume content of the PFMSC fibers is 1.0%, the response surface model predicts that when the number of freeze–thaw times is 96 and the fiber length is 9.1 mm, the compressive strength and splitting tensile strength of PFMSC are both at maximum values of 41.21 MPa and 3.2 MPa, respectively, indicating that PFMSC can maintain a good antifreeze performance within 96 freeze–thaw cycles. The desirability is 0.826, indicating that the model has high prediction reliability.

## 5. Micromorphology Analysis

Scanning electron microscopy (SEM) using a Czech TESCAN MIRA LMS was used to observe the microscopic morphologies of manufactured sand concrete specimens without PPF and PFMSC specimens, before and after the freeze–thaw cycle test. The microscopic morphologies of the PFMSC0–0 specimen and the PFMSC specimen before and after freezing and thawing are shown in Figure 12. After freezing and thawing of PFMSC0–0, the matrix pores increase and increase, and deep cracks appear; the surface of the PFMSC specimen before the freezing and thawing test is relatively smooth and flat, with fewer cracks and pores. More cracks and pores appear on the surface of the matrix after the freeze–thaw test, but because PPF belongs to the class of bundled monofilament organic fibers, it is distributed in three-dimensional random directions in concrete, and a network-like reinforcement system is formed inside it. The cracking of concrete cracks and the expansion of cracks caused by drying and chemical shrinkage during the cement hydration process are inhibited. After the freeze–thaw cycle, the connection between the fibers and the matrix is still relatively tight, indicating that the addition of PPF can effectively improve the frost resistance of manufactured sand concrete.

## 6. Conclusions

Before the freeze–thaw cycle, the overall compressive strength of PFNSC increased with the increase in PPF volume content and length, and the overall PFMSC showed a decreasing trend with the increase in PPF volume content and length. The compressive strength is the best when the content of PPF is 1.2%, and the compressive strength of PFMSC is the best when the length of PPF is 9 mm and the volume content of PFMSC is 1.0%. The overall trend is an increase, and the splitting tensile strength is the best when the PPF length is 12 mm and the volume content is 1%.With the increase in the number of freeze–thaw cycles, the exposed area of the internal aggregate of PFMSC gradually increased. After 50, 100, and 150 cycles, the average mass damage was 0.32%, 1.85%, and 4.5%, respectively. Mechanical properties and durability performance gradually deteriorated.The mechanical properties and antifreeze properties of PFNSC and PFMSC were comprehensively evaluated with the strength value damage variable as the index, which could better reflect the evolution law of freeze–thaw damage. When the volume content of PPF is 1.0% and the length is 6 mm and 12 mm, the compressive strength damage of the PFMSC specimen is lower; when the volume content of PPF is 1.0% and the length is 6 mm, the splitting tensile strength damage of the PFMSC specimen is higher. The strength damage of the PFMSC specimens is generally lower than that of the PFNSC specimens, and when the PPF length is the same, the volume content of 1% can better reduce the strength damage. According to the prediction results of PFNSC and PFMSC, the antifreeze performances of PFNSC and PFMSC are similar. This shows that polypropylene fibers have similar effects on PFNSC and PFMSC. It also shows that it is feasible to replace natural sand concrete with 100% artificial sand in practical engineering.The optimal performances of PFNSC and PFMSC are predicted by the RSM strength composite model. PFNSC can maintain a good antifreeze performance within 105 cycles of freezing and thawing. When the volume content of PPF is 1.2% and the length is 11.82 mm, the freezing performance is optimal, the compressive strength value is 33.8 MPa, and the split tensile strength value is 3.1 MPa. PFMSC can maintain a good antifreeze performance within 96 freeze–thaw cycles. When the volume content of PPF is 1.2% and the length is 9.1 mm, the antifreeze performance of the specimen reaches its maximum, its maximum tensile strength value is 45.8 MPa, and the split tensile strength value is 3.2 MPa.

## Figures and Tables

**Figure 1 polymers-14-04054-f001:**
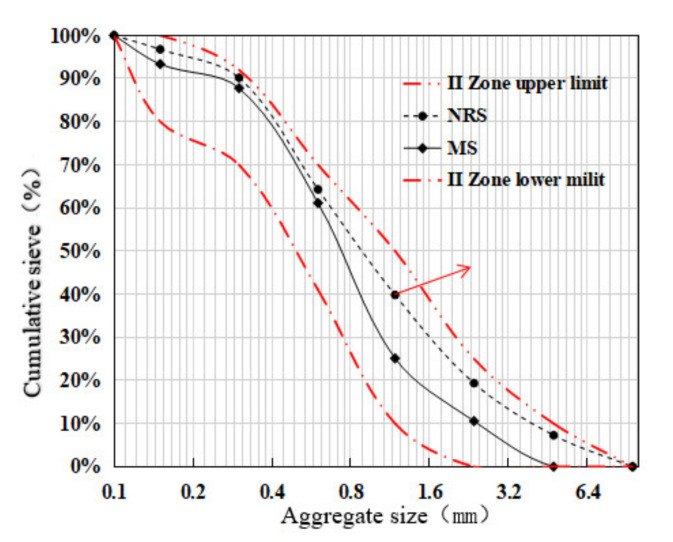
Cumulative sieve residue of fine sand.

**Figure 2 polymers-14-04054-f002:**
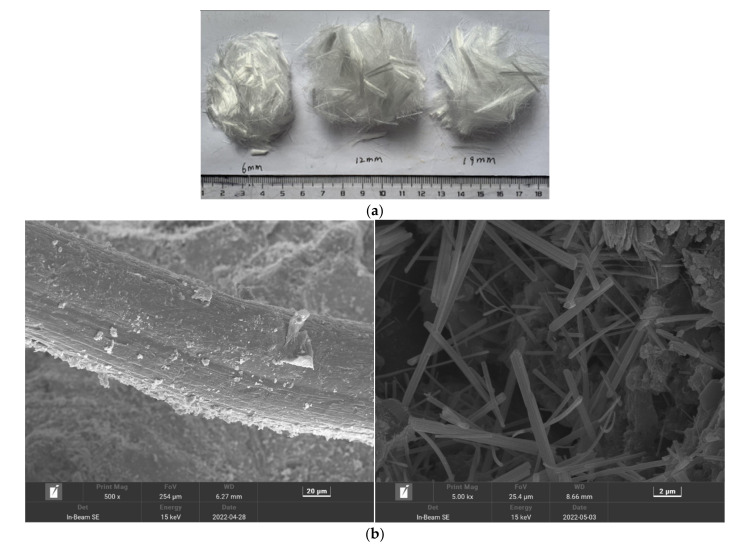
Physical and microscopic pictures of polypropylene fibers: (**a**) physical map, (**b**) micrograph.

**Figure 3 polymers-14-04054-f003:**
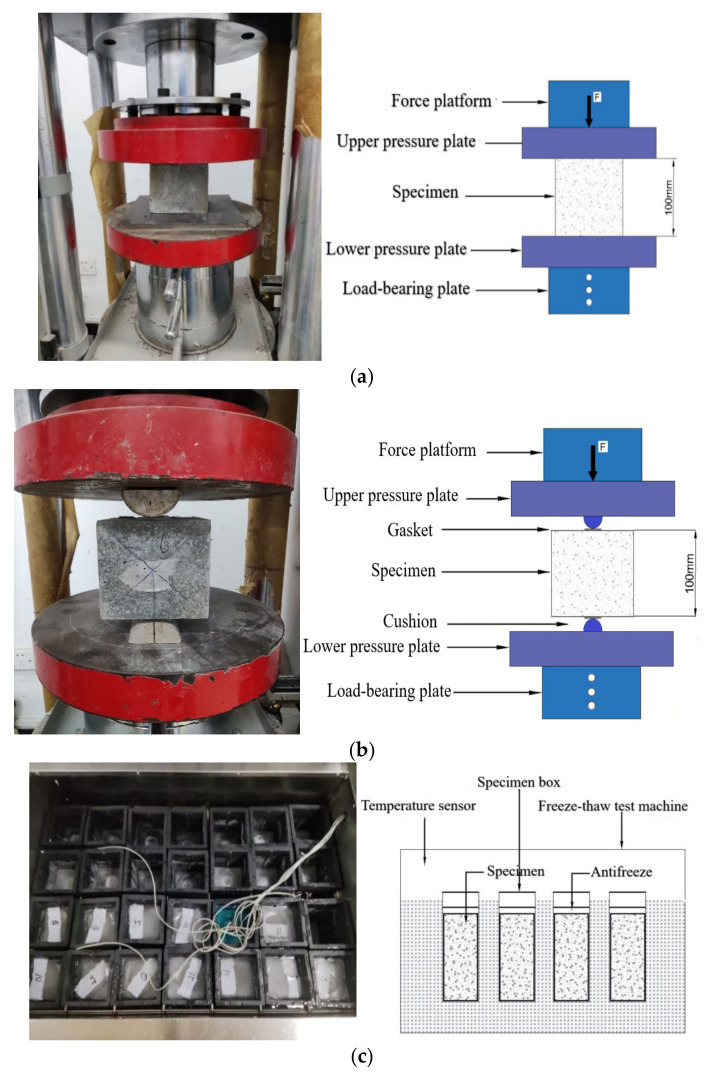
Testing device: (**a**) Compression test, (**b**) Flexural strength test, (**c**) Freeze–thaw cycle test.

**Figure 4 polymers-14-04054-f004:**
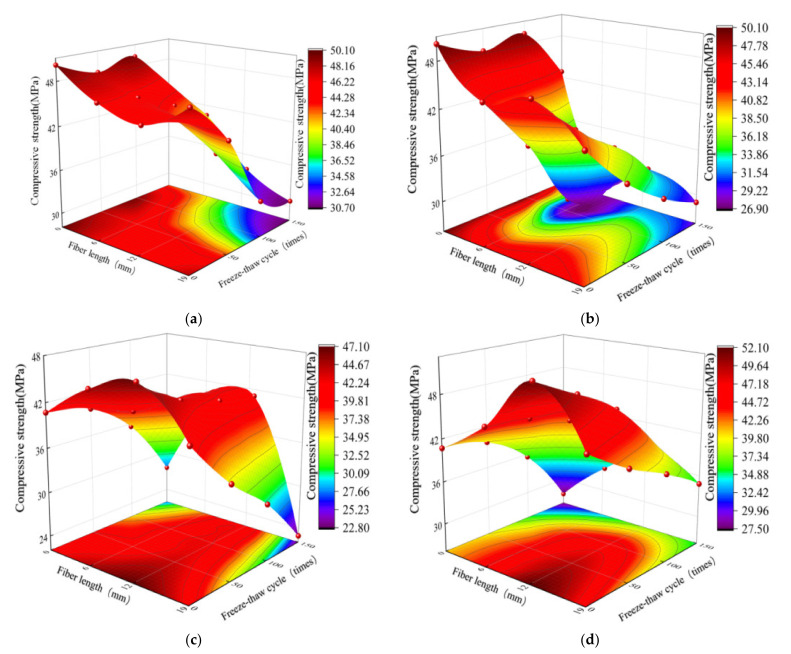
Compressive strength test value at different freeze–thaw cycles: (**a**) PFMSC compressive strength value when PPF volume is 1.0%, (**b**) PFMSC compressive strength value when PPF volume is 1.2%, (**c**) PFNSC compressive strength value when PPF volume is 1.0%, (**d**) PFNSC compressive strength value when PPF volume is 1.2%.

**Figure 5 polymers-14-04054-f005:**
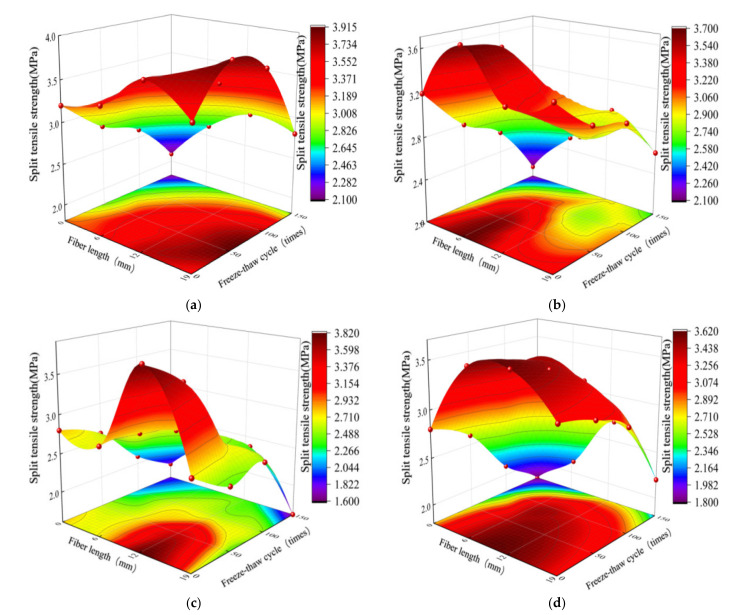
Split tensile strength at different freeze–thaw cycles: (**a**) PFMSC splitting tensile strength value when PPF volume is 1.0%, (**b**) PFMSC splitting tensile strength value when PPF volume is 1.2%, (**c**) PFNSC splitting tensile strength value when PPF volume is 1.0%, (**d**) PFNSC splitting tensile strength value when PPF volume is 1.2%.

**Figure 6 polymers-14-04054-f006:**
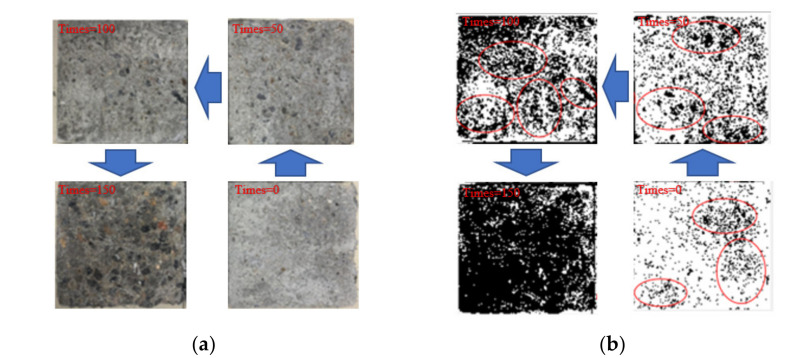
Binarization processing of specimen images with different freezing and thawing times. (**a**) Concrete apparent structure before and after freeze and thawing. (**b**) Binary treatment diagram before and after freeze and thawing.

**Figure 7 polymers-14-04054-f007:**
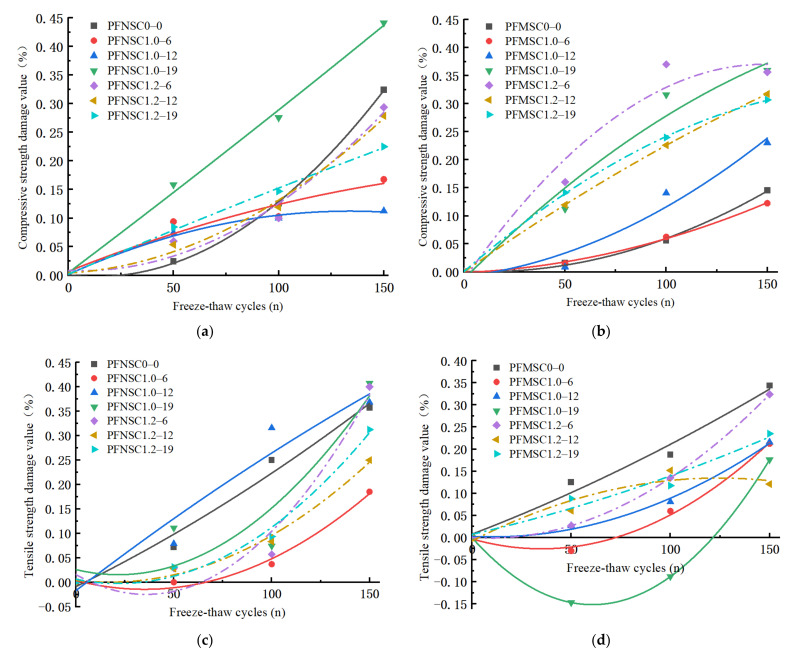
Strength damage under different freeze–thaw cycles: (**a**) PFNSC compressive strength damage rate, (**b**) PFMSC compressive strength damage rate, (**c**) PFNSC splitting tensile strength damage rate, (**d**) PFMSC splitting tensile strength damage rate.

**Figure 8 polymers-14-04054-f008:**
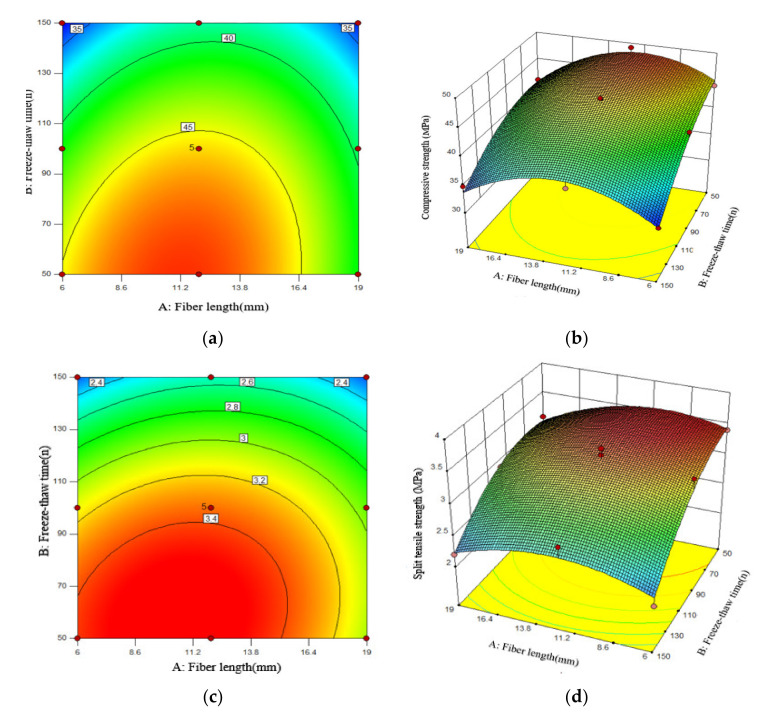
Effects of Factor A, B, and their interactions on R.(PFNSC): (**a**) Surface diagrams of compressive strength variation with A and B (2D), (**b**) Surface diagrams of compressive strength variation with A and B (3D), (**c**) Surface diagrams of split tensile strength variation with A and B (2D), (**d**) Surface diagrams of split tensile strength variation with A and B (3D).

**Figure 9 polymers-14-04054-f009:**
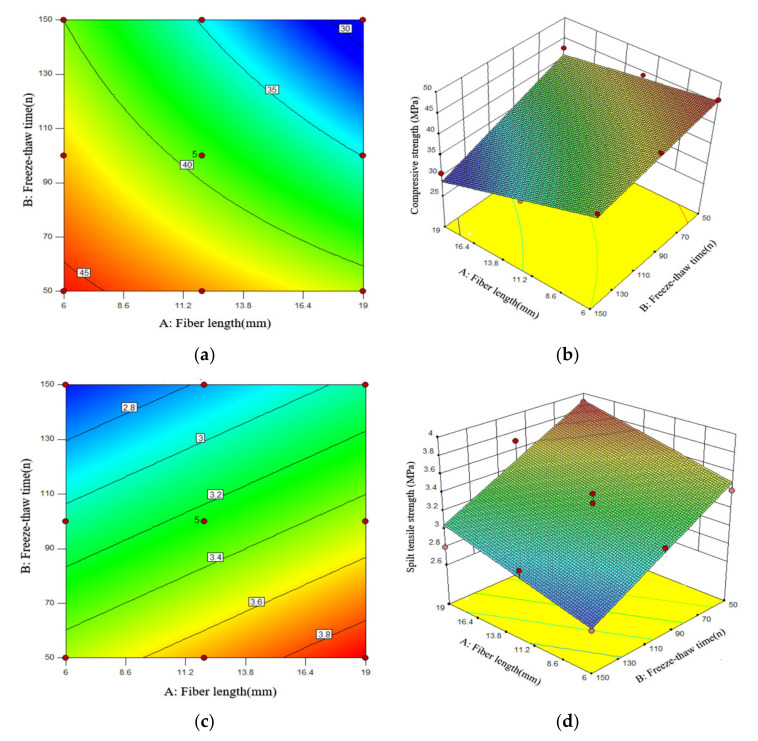
Effects of Factor A, B, and their interactions on R.(PFMSC): (**a**) Surface diagrams of compressive strength variation with A and B (2D), (**b**) Surface diagrams of compressive strength variation with A and B (3D), (**c**) Surface diagrams of split tensile strength variation with A and B (2D), (**d**) Surface diagrams of split tensile strength variation with A and B (3D).

**Figure 10 polymers-14-04054-f010:**
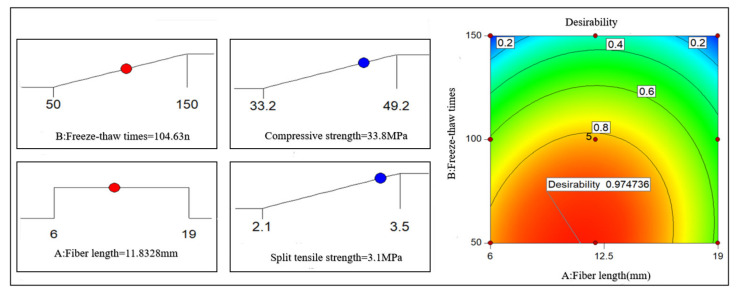
PFNSC and optimization results of strength maximization.

**Figure 11 polymers-14-04054-f011:**
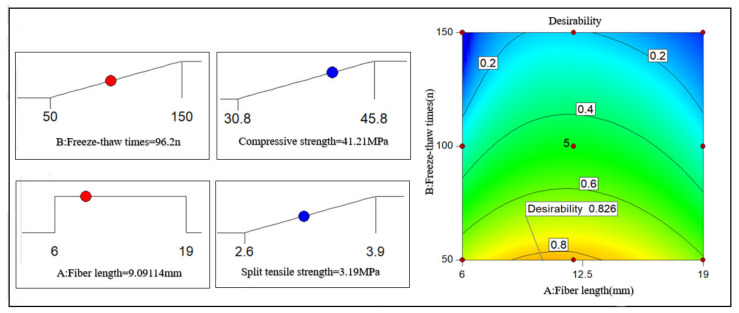
PFMSC and Optimization results of strength maximization 20.

**Figure 12 polymers-14-04054-f012:**
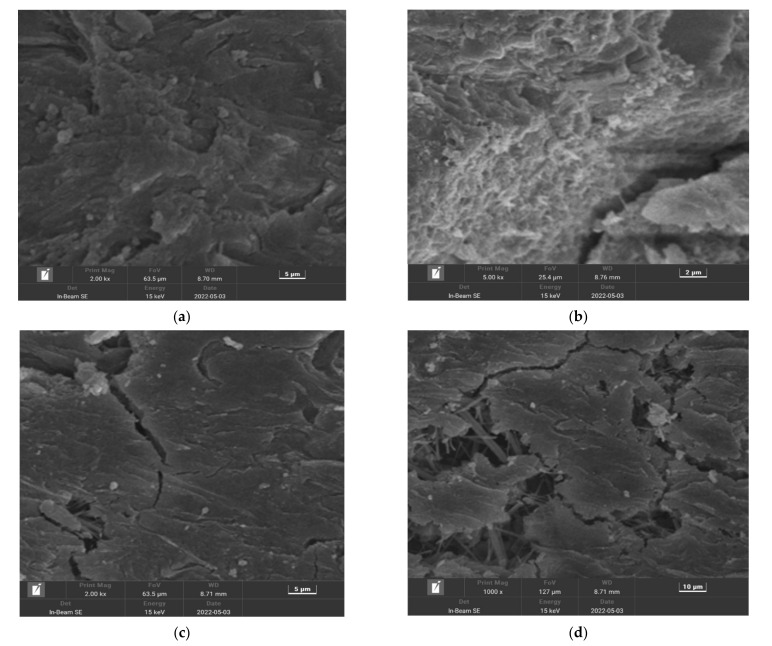
PFMSC microtopography of PFMSC before and after freeze–thawing: (**a**) PFMSC0–0: Before freezing and thawing, (**b**) PFMSC0–0: After freezing and thawing, (**c**) PFMSC1.0–9: Before freezing and thawing, (**d**) PFMSC1.0–9: After freezing and thawing.

**Table 1 polymers-14-04054-t001:** Physical indicators of polypropylene fiber.

Type	Density (g/cm^3^)	Elastic Modulus (MPa)	Break Strength (MPa)	Elongation at Break (%)	Melting Point (°C)	Ignite (°C)	Acid and AlkaliResistance
Fascicular	0.91	>4500	450	255±	165–175	590	Better

**Table 2 polymers-14-04054-t002:** Laboratory mix proportion of concrete.

NO.	Polypropylene Fiber		Material Dosage (kg/m^3^)						
	Volume Dosage (%)	Length (mm)	Cement	Flash	Natural River Sand	Mechanism Sand	Coarse Aggregate	Water	Water Reducer
PFNSC0–0	0	0	398	80	763	0	920	200	4.71
PFNSC1.0–b	1.0	6/12/19	398	80	763	0	920	200	4.71
PFNSC1.2–b	1.2	6/12/19	398	80	763	0	920	200	4.71
PFMSC0–0	0	0	398	80	0	763	920	200	4.71
PFMSC1.0–b	1.0	6/12/19	398	80	0	763	920	200	4.71
PFMSC1.2–b	1.2	6/12/19	398	80	0	763	920	200	4.71

**Table 3 polymers-14-04054-t003:** Compressive strength, split tensile strength, and quality under different freeze–thaw cycles.

NO.	0 Freeze–Thaw Cycles			50 Freeze–Thaw Cycles			100 Freeze–Thaw Cycles			150 Freeze–Thaw Cycles		
	Strength (MPa)		Mass (kg)	Strength (MPa)		Mass (kg)	Strength (MPa)		Mass (kg)	Strength (MPa)		Mass (kg)
	Compression	Tensile		Compression	Tensile		Compression	Tensile		Compression	Tensile	
PFNSC0–0	40.7	2.8	2.405	39.7	2.6	2.402	35.6	2.1	2.398	27.5	1.8	2.381
PFNSC1.0–6	44.7	2.7	2.379	40.5	2.7	2.752	40.1	2.6	2.362	37.2	2.2	2.351
PFNSC1.0–12	47.1	3.8	2.382	43.6	3.5	2.375	42.4	2.6	2.360	41.8	2.4	2.347
PFNSC1.0–19	41	2.7	2.378	34.5	2.4	2.374	29.7	2.5	2.278	22.9	1.6	2.252
PFNSC1.2–6	47.7	3.5	2.382	44.2	3.4	2.381	42.3	3.3	2.326	33.2	2.1	2.261
PFNSC1.2–12	52	3.6	2.318	49.2	3.5	2.312	45.8	3.3	2.281	37.5	2.7	2.245
PFNSC1.2–19	44.9	3.2	2.408	41.1	3.1	2.389	38.3	2.9	2.342	34.8	2.2	2.294
PFMSC0–0	50.1	3.2	2.392	48.1	2.8	2.384	49.3	2.6	2.364	42.8	2.1	2.310
PFMSC1.0–6	46.5	3.3	2.333	45.8	3.4	2.323	43.6	3.1	2.317	40.8	2.6	2.247
PFMSC1.0–12	44.7	3.7	2.353	44.3	3.6	2.347	38.4	3.4	2.308	34.4	2.9	2.120
PFMSC1.0–19	48.1	3.4	2.391	42.7	3.9	2.383	32.9	3.7	2.293	30.8	2.8	2.105
PFMSC1.2–6	44.3	3.7	2.363	37.2	3.6	2.348	27.9	3.2	2.290	28.5	2.5	2.278
PFMSC1.2–12	46	3.3	2.397	40.5	3.1	2.394	35.9	2.8	2.381	31.4	2.9	2.294
PFMSC1.2–19	41.7	3.4	2.316	35.8	3.1	2.304	31.7	3.0	2.282	28.9	2.6	2.216
Variance	3.15	0.34	0.03	4.18	0.42	0.10	6.03	0.42	0.04	5.65	0.39	0.08

**Table 4 polymers-14-04054-t004:** Fit function of compressive strength damage value.

**PFNSC0–0**	**PFNSC1.0–6**	**PFNSC1.0–12**	**PFNSC1.0–19**
y = 2E − 5x^2^ − 0.0005x + 0.0011 R^2^ = 0.9996	y = 3E − 6x^2^ + 0.0015x + 0.007 R^2^ = 0.9307	y = 6E − 6x^2^ + 0.0016x + 0.0018 R^2^ = 0.9914	y = 7E − 7x^2^ + 0.0028x + 0.0045 R^2^ = 0.9961
**PFNSC1.2–6**	**PFNSC1.2–12**	**PFNSC1.2–19**	
y = E − 5x^2^ + 0.0001x + 0.0092 R^2^ = 0.9663	y = E − 5x^2^ + 0.0002x + 0.0041 R^2^ = 0.9996	y = 7E − 7x^2^ + 0.0016x + 0.0019 R^2^ = 0.9974	
**PFMSC0–0**	**PFMSC1.0–6**	**PFMSC1.0–12**	**PFMSC1.0–19**
y = 9E − 6x^2^ − 0.0005x + 0.0056 R^2^ = 0.9921	y=5E − 6x^2^ + 0.0002x − 0.001 R^2^ = 0.9979	y = 8E − 6x^2^ + 0.0004x + 0.0083 R^2^ = 0.9628	y = 7E − 6x^2^ + 0.0036x + 0.0127 R^2^ = 0.9634
**PFMSC1.2–6**	**PFMSC1.2–12**	**PFMSC1.2–19**	
y = 2E − 5x^2^ − 0.0056x + 0.0163 R^2^ = 0.9597	y = 6E − 6x^2^ − 0.0028x + 0.0011 R^2^ = 0.9999	y = 7E − 6x^2^ − 0.0032x + 0.0006 R^2^ = 0.9986	

**Table 5 polymers-14-04054-t005:** Fit function of split-pull strength damage values.

**PFNSC0–0**	**PFNSC1.0–6**	**PFNSC1.0–12**	**PFNSC1.0–19**
y = 4E − 6x^2^ + 0.002x − 0.0089 R^2^ = 0.9801	y = E5 x^2^ − 0.001x + 0.0037 R^2^ = 0.9882	y = 3E − 6x^2^ + 0.0031x − 0.0171 R^2^ = 0.939	y = 2E − 5x^2^ + 0.001x + 0.0259 R^2^ = 0.8601
**PFNSC1.2–6**	**PFNSC1.2–12**	**PFNSC1.2–19**	
y = 3E − 5x^2^ + 0.0023x + 0.0157 R^2^ = 0.953	y = E − 5x^2^ + 0.0005x + 0.0042 R^2^ = 0.9908	y = 2E − 5x^2^ + 0.008x + 0.0062 R^2^ = 0.9869	
**PFMSC0–0**	**PFMSC1.0–6**	**PFMSC1.0–12**	**PFMSC1.0–19**
y = 3E − 6x^2^ + 0.0017x + 0.0078 R^2^ = 0.9801	y = 2E − 5x^2^ − 0.0013x − 0.003 R2 = 0.9952	y = E − 5x^2^ − 0.0002x + 0.0027 R^2^ = 0.9947	y = 4E − 5x^2^ − 0.005x R^2^ = 0.9999
**PFMSC1.2–6**	**PFMSC1.2–12**	**PFMSC1.2–19**	
y = 2E − 5x^2^ − 0.0003x R2 = 0.9999	y = − 9E − 6x^2^ + 0.0023x − 0.0076 R^2^ = 0.9149	y = 3E − 6x^2^ + 0.0031x − 0.0073 R2 = 0.9619	

## Data Availability

The data used to support the findings of this study are available from the corresponding author upon request.

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
