# Peer review of "Effects of Polypropylene Fibers on the Frost Resistance of Natural Sand Concrete and Machine-Made Sand Concrete"

_polymers, 2022, doi:10.3390/polym14194054_

Round 1
Reviewer 1 Report
· The informal language is not suitable and should be improved extensively. The article needs major grammatical and syntax improvements. Use of English service center is recommended.
· Majority of the qualitative statements should be modified for quantified result comparisons.
· Abstract should indicate several quantitative results.
· The introduction needs to be revised for higher quality language. The authors mentioned some works without stating about the contributions, pros and cons and the how the current work would address.
· The purpose of the article should be clarified in details, why this study could be beneficent, more in depth conclusion should be provided.
· The authors mentioned “Durability is an important indicator for measuring the ability of a material to resist the long-term damage of both itself and the natural environment”. The following references should be added for comprehensiveness of this statement. 1) Compressive behavior of concrete under environmental effects. IntechOpen. 2) Temperature and humidity effects on behavior of grouts. Advances in concrete construction. 3) Nano silica and metakaolin effects on the behavior of concrete containing rubber crumbs. CivilEng. 4) Experimental investigation of sound transmission loss in concrete containing recycled rubber crumbs.
· The authors mentioned “Compared with ordinary concrete, adding fiber is one of the internationally recognized methods that can effectively reduce the brittleness and improve the toughness of concrete”. The following references should be added for comprehensiveness of this statement. 1) Investigation of steel fiber effects on concrete abrasion resistance, Advances in concrete construction.
· The freeze thaw protocols for investigating the efficiency of the fibers should be determined
· Equation used previously should be clearly referenced.
· Based on the Figure 5, more in depth conclusion could be added.
· What is the reason for having R2 values lower than 0.9 in Table 4, how the regression is less precise for some samples.
· Howe the fiber percentages are chosen? And why
· The application and limitation should be discussed
·
Author Response
Please see the attachment.An example can be found here.

Reviewer 2 Report
This manuscript studied the frost resistance of polypropylene fiber manufactured/natural sand concrete of different volumes and lengths. The results showed that with the increase in freeze-thaw cycles, the apparent structures of the fiber manufactured sand concrete gradually deteriorated, the strength decreased, and the degree of freeze-thaw damage increased. According to the strength damage model, the optimum volume of fiber for the natural sand concrete is 1.2%, and the optimum volume of fiber for the fiber manufactured sand concrete specimens is 1.0%. This manuscript needs essential modifications before it can be accepted for publication as follows:
· The English language and structure of the manuscript must be improved.
· The abstract must be revised. There is nothing called "the polypro-pylene-made sand concrete."
· The first paragraph in Sec. 2.1 must be revised.
· The apparent density of both types of sand must be provided.
· The abbreviations used in Fig. 1 must be defined in the text.
· Fig. 2.b, the scale of microscopic pictures must be added.
· Sec. 2.2.b, what is "The split-pull test was performed"!!!!? What is "split tensile strength."!!!!? What is the relation between those and Fig. 3.b.
· The number of replicas in each test must be mentioned.
· Sec. 3, is the flexural test or split tensile test made!!!!? How did this strength measure?
· The standard deviation and/or coefficient of variance must be added in Table 3
· Sec. 5, the scale of SEM pictures must be added in Fig. 12.
Author Response
Dear Editors and Reviewers:
We deeply appreciate the time and effort you’ve spent in reviewing our manuscript entitled “Research on frost resistance of polypropylene fiber manufactured sand concrete based on damage model and response surface methodology” (ID: polymers-1897520). The comments were very helpful for revising and improving our paper and provided important guidance for our research. We have studied the comments carefully and have made corrections that we hope meet with your approval. Revised portions are highlighted in yellow in the revised manuscript. Our responses to the reviewers’ comments are as follows.
Please see the attachment.
Yours Sincerely,
Junyu Long

Reviewer 3 Report
· Will a fibre, that too man made synthetic fiber get affected by frost is first question coming into my mind while reading this topic.
· Why this particular name? polypropylene fiber manufactured sand concrete? It is normal fiber impregnated concrete according to me except new name.
· Is the freeze thaw cycle decided based on literature or authors own discretion. RSM output is based on input and hence I wish it can be based on literature.
· Durability is a common word used for many instances, what way authors define it here.
· one of the internationally recognized methods – method
· Literature review is not appropriate and matching need of article, authors studied about fibre reinforcement but not freeze thaw of fibers.
· incorporating polypropylene fibers of different lengths and volumes – This is not focussing on f-t cycle but only another fiber reinforcement study, f-t cycle variation based on material strength and not proportion of fiber used.
· Fig 2 shows normally used fiber and its another fibre reinforced concrete work. Ex: Material Properties of Synthetic Fiber–Reinforced Concrete under Freeze-Thaw Conditions | Journal of Materials in Civil Engineering | Vol 30, No 6 (ascelibrary.org)
· How table 1 properties are calculate, is it secondary data?
· Authors should define the novelty of the work and how it differ from other work in introduction itself, this will make readers to understand the concept quickly.
· F-T test process adopted is good. But what consists of a cycle need to be defined clearly.
· When we speak about durability, why authors refrained them only with compression and tension test is not clear. Atleast NDT, carbonation tests if done might have shown some good results, this is my suggestion, not need to incorporate in revision.
· Is section 4 necessary? Already the data is sufficient, this particular portion has only secondary data and not providing any real meaning. Performing regression analysis without proper statistical interpretation wont add any value, authors may consider.
· In my opinion section 5 has more validity in this work, please focus on that
Author Response
Dear Editors and Reviewers:
We deeply appreciate the time and effort you’ve spent in reviewing our manuscript entitled “Research on frost resistance of polypropylene fiber manufactured sand concrete based on damage model and response surface methodology” (ID: polymers-1897520). The comments were very helpful for revising and improving our paper and provided important guidance for our research. We have studied the comments carefully and have made corrections that we hope meet with your approval. Revised portions are highlighted in yellow in the revised manuscript. Our responses to the reviewers’ comments are as follows.
Please see the attachment.
Thanks and Best regards!
Yours Sincerely,
Junyu Long

Round 2
Reviewer 1 Report
The purpose of the article should be clarified in details, why this study could be beneficent, more in depth conclusion should be provided.
· The authors mentioned “Durability is an important indicator for measuring the ability of a material to resist the long-term damage of both itself and the natural environment”. The following references should be added for comprehensiveness of this statement. 1) Compressive behavior of concrete under environmental effects. IntechOpen. 2) Temperature and humidity effects on behavior of grouts. Advances in concrete construction. 3) Nano silica and metakaolin effects on the behavior of concrete containing rubber crumbs. CivilEng. 4) Experimental investigation of sound transmission loss in concrete containing recycled rubber crumbs.
Although the authors mentioned that refs are added; however the reference list is not updated. The refs should be precisely updated in the reference list
Reviewer 2 Report
The authors have successfully addressed all my comments. Therefore, I recommend the publication of this manuscript.
Reviewer 3 Report
Dear Author,
I had gone through the comments and revisions done, I am satisfied with the revisions, please proceed.